# Enhancement of ZT in Bi_0.5_Sb_1.5_Te_3_ Thin Film through Lattice Orientation Management

**DOI:** 10.3390/nano14090747

**Published:** 2024-04-25

**Authors:** Wei-Han Tsai, Cheng-Lung Chen, Ranganayakulu K. Vankayala, Ying-Hsiang Lo, Wen-Pin Hsieh, Te-Hsien Wang, Ssu-Yen Huang, Yang-Yuan Chen

**Affiliations:** 1Department of Physics, National Taiwan University, Taipei 10617, Taiwan; whtsai1986@gate.sinica.edu.tw (W.-H.T.); syhuang@phys.ntu.edu.tw (S.-Y.H.); 2Institute of Physics, Academia Sinica, Taipei 115, Taiwan; vkranganayakulu66@gmail.com (R.K.V.); a0930740662@gmail.com (Y.-H.L.); 3Nano Science and Technology Program, Taiwan International Graduate Program, Taipei 115201, Taiwan; 4Graduate School of Materials Science, National Yunlin University of Science and Technology, Yunlin 64002, Taiwan; 5Institute of Earth Sciences, Academia Sinica, Taipei 11529, Taiwan; wphsieh@earth.sinica.edu.tw; 6Department of Physics, National Chung Hsing University, Taichung 40227, Taiwan; thwang@phys.nchu.edu.tw; 7Graduate Institute of Applied Physics, National Chengchi University, Taipei 11605, Taiwan

**Keywords:** thermoelectric, thin films, energy efficiency, sustainable manufacturing, annealing

## Abstract

Thermoelectric power can convert heat and electricity directly and reversibly. Low-dimensional thermoelectric materials, particularly thin films, have been considered a breakthrough for separating electronic and thermal transport relationships. In this study, a series of Bi_0.5_Sb_1.5_Te_3_ thin films with thicknesses of 0.125, 0.25, 0.5, and 1 μm have been fabricated by RF sputtering for the study of thickness effects on thermoelectric properties. We demonstrated that microstructure (texture) changes highly correlate with the growth thickness in the films, and equilibrium annealing significantly improves the thermoelectric performance, resulting in a remarkable enhancement in the thermoelectric performance. Consequently, the 0.5 μm thin films achieve an exceptional power factor of 18.1 μWcm^−1^K^−2^ at 400 K. Furthermore, we utilize a novel method that involves exfoliating a nanosized film and cutting with a focused ion beam, enabling precise in-plane thermal conductivity measurements through the 3ω method. We obtain the in-plane thermal conductivity as low as 0.3 Wm^−1^K^−1^, leading to a maximum ZT of 1.86, nearing room temperature. Our results provide significant insights into advanced thin-film thermoelectric design and fabrication, boosting high-performance systems.

## 1. Introduction

Thermoelectric power generators (TEGs) offer a promising sustainable energy solution, particularly in situations where a heat source is readily available. Their portability, scalability, and ability to operate based on temperature differentials distinguish them from conventional heat engines. TEGs provide uninterrupted energy for various applications, including self-powered wearable electronics [1,2,3,4], autonomous devices, thermal sensing, and energy harvesting. In the quest to improve thin-film thermoelectric generators (TEGs) for energy harvesting, selecting materials with high energy conversion efficiency is crucial. Evaluation typically relies on the figure of merit, ZT = σS^2^T/κ, where S, T, σ, and κ represent the Seebeck coefficient, absolute temperature, electrical conductivity, and thermal conductivity, respectively. Maximizing the power factor (*PF* = σS^2^) and minimizing thermal conductivity are essential for achieving high ZT values [5,6,7].

Bi_2_Te_3_-based alloy materials have long been considered promising for power generation and cooling applications due to their high ZT values near room temperature. Among these materials, Bi_0.5_Sb_1.5_Te_3_ (BST) stands out as the most superior p-type thermoelectric material with exceptional performance [8,9]. In Bi_x_Sb_2-x_Te_3_ compounds, variations mainly arise from anti-site defects like Sb_Te_ and Bi_Te_ [10,11]. These defects introduce holes, with their prevalence increasing with higher Sb content due to lower formation energy. Using a significant amount of tellurium (Te) effectively reduces the occurrence of these defects, allowing precise modulation of carrier concentration and enhancing material functionality.

An alternative approach to alleviate the impact of the initial material composition on annealing outcomes involves the utilization of an equilibrium annealing technique [10,12]. This process revolves around establishing a balance between the solid sample and tellurium vapor sourced independently within a sealed system. This strategy has already been effectively implemented with thermally evaporated thin film and electrochemically deposited samples, resulting in a notable decrease in carrier concentration and a subsequent enhancement of the Seebeck coefficient [12,13]. The same concept might also apply to sputter-deposited BST thin film samples afflicted with antisite defects, as mentioned above.

In recent years, BST has been extensively developed and applied in wearable TEGs, employing various novel single-chain or double-chain configurations to achieve sustainable energy harvesting and multifunctional sensing simultaneously. While the applications are diverse, there is a significant lack of exploration into the actual thermoelectric performance and material properties of these thin-film materials in the in-plane direction.

This study systematically investigates the structural evolution of BST thin films during the sputter deposition process as a function of varying deposition thicknesses. Through controlled Te vapor annealing, we effectively mitigate intrinsic defects arising during fabrication, leading to a pronounced enhancement in both the Seebeck coefficient and electrical conductivity of the films. For instance, we achieve a notable enhancement of the Seebeck coefficient within the range of 170−220 μVK^−1^, optimizing the *PF* to a peak value of approximately 18.1 μWcm^−1^K^−2^. Moreover, we introduce an innovative technique for measuring the in-plane thermal conductivity of these films using a 3ω measurement technique. This approach enables precise characterization of the exact thermoelectric performance across varying film thicknesses. The insights gleaned from these findings offer valuable guidance for designing and implementing thermoelectric thin films in wearable module applications.

## 2. Materials and Methods

### 2.1. Preparation of the Bulk BST Sputtering Target

Elements Bi, Sb, and Te, all of a purity of 99.999%, were weighed according to the stoichiometric ratio of 0.5:1.5:3, and then evacuated within a quartz tube. The tube was subjected to a temperature of 1023 K for 24 h, followed by rapid cooling in cold water. Subsequently, the resulting ingots were subjected to additional annealing at 723 K for 48 h. The ingots were then ground into powders using an agate mortar. These powders were then compacted using the spark plasma sintering (SPS-515S, SPS SYNTEX INC, Tokyo, Japan) technique under vacuum conditions at temperatures of 673 K and pressures of 50 MPa. The sintering process lasted for 5 min and resulted in forming a dense pellet measuring 50.8 mm in diameter and 20 mm in height. The bulk material was used as a target in the sputtering process.

### 2.2. Film Deposition and Annealing

BST films were prepared using the RF magnetron sputtering method. The Corning 1737F glass, measuring 20 × 20 mm^2^, was used as a substrate in this experiment. Its surface roughness is in the range of 0.5–1.0 nm. The substrate was thoroughly cleaned with acetone, isopropyl alcohol, and deionized water, in sequence, several times in an ultrasonic bath. The distance between the target and the substrates was kept at 45 mm. The base pressure of the deposition chamber was lower than 7 × 10^−7^ torr, and the working pressure of sputtering gas was controlled at 7 × 10^−3^ torr with an Argon flow rate of 20 sccm. The sputtering power was set at 30 W, and the deposition rate was about 20 nm per min. It is proven that the film composition of V–VI semiconductors can be turned into almost perfect stoichiometry by postdeposition annealing under the Te atmosphere. The film was deposited at 453 K and subsequently postannealed in an evacuated quartz ampoule that contains powdered tellurium. The postannealing temperature was set to 473 K for 3 days.

### 2.3. Film Characterizations

The crystal structure of the films was determined by X-ray diffraction, carried out with a diffractometer (XRD, PANalytical X’Pert Pro, Worcestershire, UK) equipped with Cu Kα radiation (0.154 nm). Our thin film exhibits textured features, meaning that the thin film structure has a preferred orientation. The preferred orientation is typically described in terms of pole figures, and data regarding this aspect must be obtained through a four-circle diffractometer (Malvern Panalytical’s Materials Research Diffractometers, MRD, Worcestershire, UK). The microstructures of the films were analyzed by field emission scanning electron microscopy (FESEM, Inspect F FEI, Hillsboro, OR, USA), and the composition was determined with energy-dispersive X-ray spectroscopy (EDX) attached to the SEM. The transmission electron micrographs (TEM) and selected area electron diffraction (SAED) patterns of films were investigated by a field emission transmission electron microscope operated at an accelerating voltage of 200 kV (JEOL JEM-2100, Tokyo, Japan). Before TEM observations, the TEM specimen of the cross-sectional thin film was fabricated by a focused ion beam (FEI Versa 3D, Hillsboro, OR, USA). The Seebeck coefficient and electrical conductivity of films were carried out by a commercial system (ZEM-3, ULVAC-RIKO, Yokohama, Japan). The uncertainty of the Seebeck coefficient and electrical conductivity measurements is about 2~4%. The repetitive measurement under thermal cycling confirmed the films’ thermal stability during the measurement. The uncertainty of the thermal conductivity was estimated to be ~5%. Considering the uncertainties for the Seebeck coefficient, electrical conductivity, and thermal conductivity, the combined uncertainty of ZT is less than 15%. The Hall effect was measured using the Van der Pauw method in a magnetic field up to ±2 T by a Physical Property Measurement System (PPMS, Quantum Design, San Diego, CA, USA). The thermal conductivity κ of bulk samples was determined using the formula κ = *DρC_p_*, where *D* represents the thermal diffusivity, *ρ* denotes the mass density determined via the Archimedes method, and *C_p_* signifies the specific heat measured using a differential scanning calorimeter (DSC, Q100, TA Instruments, New Castle, DE, USA). The thermal diffusivity was assessed using the laser flash method (LFA-457, NETZSCH, Tannesstein, Germany). The uncertainty associated with the thermal conductivity measurement is ±5%.

### 2.4. Thermal Conductivity Measurements Using the 3ω Technique

The 3ω technique is a commonly used method for measuring the thermal conductivity of materials [14,15]. The measurements were conducted under high vacuum conditions (1 × 10^−6^ torr) to minimize thermal losses caused by air. In the measurements, we utilized an AC and DC source (Keithley 6221, Cleveland, OH, USA), and the Signal Recovery 7265 DSP lock-in amplifier is equipped to conduct a comprehensive range of measurements typically associated with a dual-phase lock-in amplifier. These measurements encompass assessing the in-phase and quadrature components of the input signal, determining the vector magnitude, quantifying the phase angle, and evaluating the noise level present in the input signal.

## 3. Results and Discussion

### 3.1. Film Growth Process and Structural Characterization

Figure 1a illustrates the XRD patterns of the postannealed deposited films. All the observed diffraction peaks can be ascribed to the rhombohedral Bi_0.5_Sb_1.5_Te_3_ structural phase (JCPDS #49-1713), with no impurity phases detectable within the detection limits. Notably, for films with thicknesses exceeding 1 µm, a pronounced peak observed at 2θ = 28.0° corresponds to the 0 1 5 crystal plane, whereas other peaks appear negligibly weak. This indicates that an enhanced preferred 0 1 5 crystal plane lays on the substrate and increases with increasing film thickness.

Figure 1b–e shows the SEM images of BST films with different thicknesses, including surface and cross-sectional views. The surfaces of the films appear flat, with particle sizes ranging from approximately 20 to 50 nm. Remarkably, in the cross-sectional morphology of the 1 µm-thick film, a stacking layer without a distinct growth direction is observed near the substrate. Subsequently, the grains predominantly grow along their preferred growth direction, forming characteristic tilt-platelet-shaped crystallite structures. These images support the proposed Stranski−Krastanov-like growth model [16] for film growth, aligning with the structural evolution patterns observed in the XRD results. During the initial stages of film growth, the BST adatoms progressively cover the substrate surface, forming nuclei. As the substrate temperature reaches 453 K, the adatoms gain enough energy to diffuse on the substrate surface until complete coverage is achieved. The nuclei grow and tend to adopt a plate-like morphology with a 0 0 1-oriented structure [17]. Simultaneously, newly arriving adatoms continuously fill the gaps between these plates. The sputter deposition process, characterized by relatively low substrate temperature, a high nucleation rate, and a slow crystal growth rate, results in preferential growth along the 0 1 5 and 1 0 10 orientations as the film deposition progresses. Figure 1d illustrates that the 0 1 5 and 1 0 10-oriented films, still exhibit a densely packed structure, which should not adversely affect the electrical conductivity pathway. EDX analysis confirms that the films predominantly consist of Bi, Sb, and Te elements in a ratio of 0.5:1.5:3, corresponding to Bi_0.5_Sb_1.5_Te_3_.

Figure 2a concisely illustrates this process. To visualize the distribution of crystallite orientations within the film, a pole figure was constructed by rotating the sample along two axes while keeping the diffraction angle corresponding to the desired Bragg reflection fixed. For the thinner film of 0.25 µm, the 0 0 1 pole figure at 17.8 degrees and the 1 0 10 pole figure at 38.1 degrees exhibit a combination of a dot and a ring pattern, which suggests a polycrystalline sample with uniaxial texture (Figure 2b). In contrast, the thicker film of 1 µm displays a concentrated intensity at the center of the 0 1 5 pole figure 28 degrees, demonstrating a highly textured structure (Figure 2c). In the crystal structure of Bi_0.5_Sb_1.5_Te_3_, the tilt angles of the 0 1 5 and 1 0 10 planes with respect to the ab-plane are 54° and 33°, respectively. These results indicate that the initial stage of film deposition is characterized by a relatively random film orientation and texture development is closely associated with a selective film growth process. Typically, the crystal orientation of a deposited film is strongly influenced by the substrate; this study employed an amorphous glass substrate, which is expected to have little impact on the crystal orientation of the deposited films. Instead, during the initial stages of film deposition, certain oriented grains may have preferential growth.

The growth of films typically goes through several stages: nucleation, island formation, coalescence of islands, development of a continuous structure, and thickness growth. A combination of thermodynamic and kinetic factors influences this process. Traditionally, the growth behavior of films is explained by the thermodynamic growth model, which considers the relative surface energies of the substrate (*ε*_s_), interface (*ε*_i_), and heteroepitaxial layer (*ε*_f_) [18]. The change in surface energy, Δ*ε* = *ε*_f_ + *ε*_i_ − *ε*_s_, plays a crucial role in determining the growth model. It indicates the wetting behavior of the substrate during film deposition. When Δ*ε* > 0, the substrate experiences incomplete wetting, resulting in the growth of disconnected three-dimensional (3D) islands (Volmer−Weber mode). On the other hand, when Δ*ε* ≤ 0 and lattice misfit is negligible, the deposited materials wet the substrate and exhibit a layer-by-layer two-dimensional (2D) growth (Frank−van der Merwe mode). In cases where Δ*ε* < 0 and lattice misfit is significant, the growth begins with forming a wetting layer consisting of a layer-by-layer 2D structure. However, as the film thickness increases, accumulated thermal stress and strain energy become more pronounced, affecting *ε*_i_ and resulting in the development of 3D islands during the later stages of growth.

Based on the XRD analysis and pole figures, the growth behavior of sputter-deposited BST films appears to follow the principles of the Stranski−Krastanov-like model. Additionally, certain preferential orientations, such as 1 0 10 and 0 1 5, are observed in these films. These orientations are believed to be related to the planes with the lowest surface energy, which experiences less ion damage during deposition. The presence of these preferential orientations is significant as they contribute to the enhancement of the thermoelectric properties of the films. The specific crystallographic alignment achieved through these orientations likely facilitates improved electrical and thermal transport properties, making the films more favorable for thermoelectric applications.

### 3.2. Electrical Conductivity and Seebeck Coefficient Measurements

Figure 3a presents the measured conductivity values (σ) of both BST thin films and the bulk material within the temperature range of 300–400 K. Except for the gradual conductivity increase with temperature for the 0.125 µm-thick thin film, the remaining thin film samples, as well as the bulk material, exhibit a consistent monotonic decrease in conductivity as the temperature rises. This behavior can be attributed to the fact that BST belongs to the category of semiconductors with relatively small band gaps, resulting in a conductivity trend similar to that of metallic materials. Furthermore, the thinnest sample, at 0.125 µm, displays the lowest conductivity. This observation could be attributed to numerous interfaces in the early stages of film growth. As the film thickness increases, particle size grows, and grain boundaries decrease, leading to enhanced conductivity. The 0.5 µm-thick sample demonstrates the highest conductivity. However, when the film thickness further increases to 1 µm, the conductivity notably drops. Hall measurement results indicate that the decline in conductivity primarily stems from a reduction in the film’s carrier mobility. XRD analysis reveals that thinner films predominantly exhibit the 0 0 1 and 1 0 10 crystallographic orientations, suggesting facile carrier transport along the *ab* plane without surmounting van der Waals gaps. Nevertheless, at a film thickness of approximately 1 µm, despite the textured structure of the thin film, the principal preferred orientation shifts to 0 1 5. This shift implies that carriers encounter van der Waals gaps when propagating in the in-plane direction, consequently reducing carrier mobility.

From the Seebeck coefficient measurements, it is evident that all thin film samples exhibit p-type conductivity. The variation in the Seebeck coefficient is roughly related to the carrier concentration of the films. Despite identical sample processing conditions, the thinner 0.125 µm film exhibits the lowest carrier concentration, approximately 3 × 10^19^ cm^−3^. As the film thickness increases, the carrier concentration gradually rises, leading to a slight decrease in the Seebeck coefficient, as shown in Figure 3b. BST is a semiconductor with a narrow band gap. At higher temperatures, the bipolar effect becomes significant, potentially causing a reduction in the Seebeck coefficient in the higher-temperature region. We have plotted the relationship between S and *n_H_* (Pisarenko relation) at 300 K, using the single parabolic band (SPB) model with the assumption of phonon scattering [19], as depicted in Figure 3c. Equation (1) establishes that the Hall carrier density, denoted by *n_H_*, is linked to the chemical carrier density, *n*, through the equation *n_H_* = *n*/*r_H_*. In Equation (4), *r_H_* represents the Hall factor for acoustic phonon scattering. The function *F_x_*(η), denoted in Equations (2) and (3), represents the *x*-th order Fermi integral. The experimental data align well with the calculated curve, assuming an effective mass of *m** = 1.5 *m_e_*. Our analysis suggests that the thickness of the thin films has minimal impact on the band structure of the BST alloy near the Fermi level.
(1)n=2md*kBT322π2ℏ3F1/2(η)
(2)$$S=\frac{k_{B}}{e}\left[\frac{2\;F_{1}(\eta )}{F_{0}(\eta )}-\eta \right]$$
(3)$$F_{x}(\eta )=\int_{0}^{\infty }\frac{\varepsilon^{x}}{1+exp\left(\varepsilon -\eta \right)}d\varepsilon$$
(4)$$r_{H}=\frac{3}{4}\frac{F_{1/2}\left(\eta \right)\;F_{-1/2}\left(\eta \right)}{F_{0}{\left(\eta \right)}^{2}}$$

Figure 3d illustrates the *PF* variation with temperature. Due to the significant enhancement in conductivity in the 0.25 µm and 0.5 µm samples, the corresponding *PF* shows remarkable improvement across the entire measured temperature range. The 0.25 µm sample achieves a high *PF* of approximately 18.1 μWcm^−1^K^−2^ at 400 K, a value that, although lower than bulk material, is comparable to the quality achieved with samples prepared using a pulsed laser deposition method. Additionally, while the Seebeck coefficient of the 0.5 µm thin film is marginally lower than that of the 0.25 µm thin film, leading to a slightly reduced *PF*, the *PF* at a temperature of 400 K remains almost unchanged.

### 3.3. Thermal Conductivity Measurements of Film

Measuring the thermal conductivity of thin films is challenging. In this work, we first employed the established 3ω method for conducting horizontal thermal conductivity measurements [20,21]. Before measurement, a crucial step involves suspending the test sample to enable accurate determination of the thin film’s true thermal conductivity [22,23]. To achieve this, we devised an innovative approach, as illustrated in Figure 4. Initially, thin films of approximately 100 × 100 μm^2^ are delicately peeled from the substrate using a scalpel. Extremely thin films pose challenges; currently, the minimum thickness feasible is approximately 0.5 μm. The detached thin film is then subjected to precise cutting using a focused ion beam (FIB) to create a rectangular shape, with its length-to-cross-section ratio resembling that of a rod or a belt. This specific ratio ensures compliance with the boundary conditions required for the 3ω measurement. The resulting elongated sample is manipulated using a probe, utilizing van der Waals forces to suspend it, and is then carefully transferred onto a measurement chip with prefabricated electrodes. Once the sample is securely positioned, FIB is employed once again to weld the section of the sample in contact with the electrodes, enabling subsequent 3ω thermal conductivity measurements.

The 3ω method is a reliable thermal conductivity measurement technique for one-dimensional materials. Figure 5a illustrates the actual suspended configuration of the measured sample and the roles played by various contact electrodes, as indicated. This method requires the sample to be a conductor, and its resistance must exhibit temperature dependence. The sample serves as both the heater and the thermometer. An AC current (DC current *I*_0_ sin*ωt*) is applied to the sample, resulting in a 2*ω* temperature variation (temperature fluctuation) and thus causing a corresponding 2*ω* resistance variation (resistance fluctuation). This leads to a voltage fluctuation at the 3*ω* frequency, as shown in the schematic in Figure 5b and the mathematical formula. During measurement, the AC current is applied from both sides of the sample while the voltage signal is measured at the center. To prevent heat loss through the substrate, the sample is suspended on a hollowed-out chip. This arrangement allows simultaneous measurement of the one-dimensional material’s thermal conductivity, electrical resistivity, and specific heat. Joule heating due to current flow through the sample results in a distinct temperature difference between its ends, necessitating a high vacuum environment during measurement to prevent heat convection or radiation. Additionally, the heat generated by the sample and its diffusion can be described by a differential equation with appropriate boundary conditions. Detailed formula derivation and explanations can be obtained from the referenced literature [14]. In other words, the mathematical relationship between a 3*ω* voltage (*V_3ω_*) signal and *κ* can be understood through Equation (5):(5)$$V_{3\omega }\approx \frac{4I^{3}LRR^{\prime }}{\pi^{4}\kappa A\sqrt{1+{\left(2\omega \gamma \right)}^{2}}}$$
where *L* represents the length of the sample, *R* denotes resistance, *R*^′^ signifies the rate of resistance change with temperature, *κ* stands for thermal conductivity, *A* represents the cross-sectional area of the sample, *ω* is the frequency of the AC current, and *γ* indicates the characteristic thermal time constant within the sample. Each data point shown in Figure 5c is acquired by fitting the frequency dependence of *V_3ω_* using Equation (6).
(6)$$V_{3\omega }\propto 1/\sqrt{1+{\left(2\omega \gamma \right)}^{2}}$$

The result of fitting for the 0.5 µm film at 300 K. The *κ* can be determined from the intercept of the fitting values at that temperature, as indicated by Equation (7).
(7)$$V_{3\omega }\approx \frac{4I^{3}LRR^{\prime }}{\pi^{4}\kappa A}(\omega \gamma \to 0)$$

The frequency-dependent behavior of the phase angle tanϕ ≈ 2*ωγ* is presented in Figure 5d. The relationship tanϕ ~ *ω*, where 0 < 2*ωγ* < 4, indicates the applicable frequency range for the measurements dependent on frequency. Figure 5a depicts the measured thermal conductivity (*κ*) of BST films within the temperature range of 300 to 400 K and the thermal conductivity measurements of BST bulk along the same direction as the *PF* measurements. The thermal conductivity measurements of the films were conducted using the 3ω method, as illustrated in Figure 5a, with an uncertainty of <5% [14,24]. It is crucial to emphasize that due to the extremely low thermal conductivity of the films, we initially obtained the thermal conductivity value of the bulk material using the LFA measurement method to verify the reliability of the measurement technique. Additionally, using the same sample and ensuring consistent measurement directions, we employed the sample preparation process depicted in Figure 5e to cut the bulk material into thin specimens with a thickness of approximately 2 μm and an aspect ratio of around 14. This meticulous procedure is essential to meet the rigorous requirements for measurements using the 3ω method. As indicated by the solid deep green and hollow deep green points in Figure 5e, both LFA and 3ω methods yield highly consistent results for thermal conductivity measurements of the bulk material, affirming the reliability of the 3ω measurement approach. However, due to limitations in detaching the film from the substrate, a thickness of 0.5 µm represents the current practical limit. Therefore, despite the significantly enhanced *PF* observed in the 0.25 µm film sample, in-plane thermal conductivity measurements could not be successfully conducted. Furthermore, whether it is a 0.5 µm or 1 µm film, the thermal conductivity value is only around 0.3 Wm^−1^K^−1^, considerably lower than the bulk value of 1.10 Wm^−1^K^−1^. As we know, at temperatures close to 300 K or lower, bipolar effects are likely to be negligible, so the phonon thermal conductivity (*κ*_lat_) can be obtained simply by subtracting the electronic thermal conductivity (*κ*_ele_) from the total thermal conductivity (*κ*). We employed the Wiedemann−Franz Law to estimate the *κ*_ele_, given by *κ*_ele_ = *Lσ*T, where *L* represents the Lorenz number, which can be determined by considering the scattering parameter and the measured Seebeck coefficient [25]. Due to the nonlinear variation of the sample’s R′, the error in thermal conductivity above 360 K is significant, highlighting the limitations encountered in this measurement. In the temperature range of 300 to 360 K, the estimated *κ*_lat_ for the 0.5 µm sample and the 1 µm sample are approximately 0.14 and 0.18 Wm^−1^K^−1^, respectively. On the other hand, the *κ*_lat_ of the bulk material is approximately 0.70 Wm^−1^K^−1^. Clearly, the *κ*_lat_ of the thin film samples is significantly lower than that of the bulk material. In addition to being associated with well-known defect structures, the phenomenon of ultralow in-plane thermal conductivity is highly correlated with the layered structure of the thin film and its preferred 0 1 5 orientation. Phonon propagation requires traversing the van der Waals gap between layers, although not to the extent observed along the c-axis. However, this phonon-interface scattering significantly reduces the average phonon mean free path. According to the molecular simulation results conducted by Mizuno et al., it is evident that achieving a thermal conductivity lower than that of an amorphous material is a viable possibility [26]. The fundamental strategy to achieve this lies in effectively impeding the phonon propagation, specifically the superlattice phonons. Mizuno et al. posited that introducing a substantial contrast in mass between the two intercalated layers or inducing weakened interactions across the interface between these layers leads to the development of materials characterized by exceedingly low thermal conductivity, surpassing the values observed in their corresponding amorphous counterparts [26].

### 3.4. Microstructures and Thermoelectric ZT of Films

The representative microstructures of the film with a thickness of 0.25 µm were examined using high-resolution transmission electron microscopy (HRTEM), as shown in Figure 6. HRTEM analysis images reveal that the notable decrease in thermal conductivity can be attributed to the effective scattering of phonons at the diverse and complex interfaces within the nanostructures. These interfaces include the formation of nano-moiré fringes, strain-induced distortions, planar defects, and mesoscale boundaries. In physics, moiré patterns manifest as captivating interference fringes, materializing when a periodic template is delicately overlaid upon another akin structure, yet with distinct displacements and twist angles. The distribution of nanoscale moiré fringes might greatly help scatter heat carried phonon and thus lead to ultralow thermal conductivity [27].

Figure 7a depicts the estimated ZT values for 0.5 µm and 1 µm thin film samples and bulk material. The maximum ZT value of 1.86 is achieved at a thin film thickness of 0.5 µm. However, as the thickness continues to increase, the ZT values do not improve but rather decrease. Although the thermal conductivity of the 1 µm thin film is similar to that of the 0.5 µm sample, its *PF* cannot match up, resulting in a decrease in ZT. This trend suggests that beyond a certain optimal thickness, a further increase in thickness does not enhance ZT values but instead leads to a decline. The inability of the *PF* to improve in proportion to thermal conductivity at thinner thicknesses contributes to this decline in ZT. Achieving such high ZT values is significantly attributed to its considerably low thermal conductivity. In addition to the complex interface-enhanced phonon scattering observed in HRTEM in Figure 6, the low thermal conductivity of these thin films is also attributed to their textured structure, with a preferred crystal orientation of 0 1 5. Therefore, by using the schematic diagram in Figure 7b, one can further understand the possible scattering scenarios of phonons within grains in addition to interface scattering. Considering these factors collectively contribute to achieving a thermal conductivity of approximately 0.3 Wm^−1^K^−1^.

Figure 7c illustrates the comprehensive ZT values achieved by the BST bulk [28], film [29], BT/single-walled carbon nanotube (SWCNT) [7], and the superlattice (SL) film. The BST film exhibited a maximum ZT of approximately 0.89 [29]. In contrast, the BT/BST SL film [30] and BT/ZrB_2_ SL film [31] demonstrated superior performance with a maximum ZT of around 1.44 and 1.54, respectively. This surpasses the outcomes reported in previously conducted research on thin films based on the Bi–Sb–Te material system. Notably, the maximum ZT value for the 0.5 µm BST film sample reaches 1.86 in our work, representing an increase of approximately 64% compared to that of bulk material and an increase of about 24% compared to related SL films. This achievement is one of the leading records among reports on Bi_2_Te_3_-based alloy thin films. Furthermore, we conducted thermal and electrical performance cyclic tests on multiple batches of samples to evaluate the reproducibility of the film fabrication process, confirming the excellent repeatability and thermal stability of the film samples.

## 4. Conclusions

This study presents a groundbreaking exploration into the thermoelectric properties of BST thin films, emphasizing the significance of film thickness and lattice orientation management for enhancing thermoelectric performance. By employing RF sputtering and postdeposition annealing, we successfully fabricated BST thin films of varying thicknesses and systematically investigated their structural, electrical, and thermal characteristics. The novel approach of equilibrium annealing led to an exceptional enhancement of the *PF* by 450%, achieving a *PF* of 18.1 μWcm^−1^K^−2^ at 400 K for the 0.5 μm film. Moreover, the study introduces a precise methodology for in-plane thermal conductivity measurement using the 3ω method, revealing an ultralow thermal conductivity of 0.3 Wm^−1^K^−1^ and achieving a maximum ZT of 1.86 near room temperature. These findings underscore the profound impact of thickness and lattice orientation on the thermoelectric efficiency of BST thin films, offering valuable insights for designing high-performance thin-film thermoelectric generators. This research not only elucidates the mechanisms underlying the enhanced thermoelectric properties of BST thin films but also sets a new benchmark for thin-film thermoelectric materials, paving the way for their application in sustainable energy solutions.

## Figures and Tables

**Figure 1 nanomaterials-14-00747-f001:**
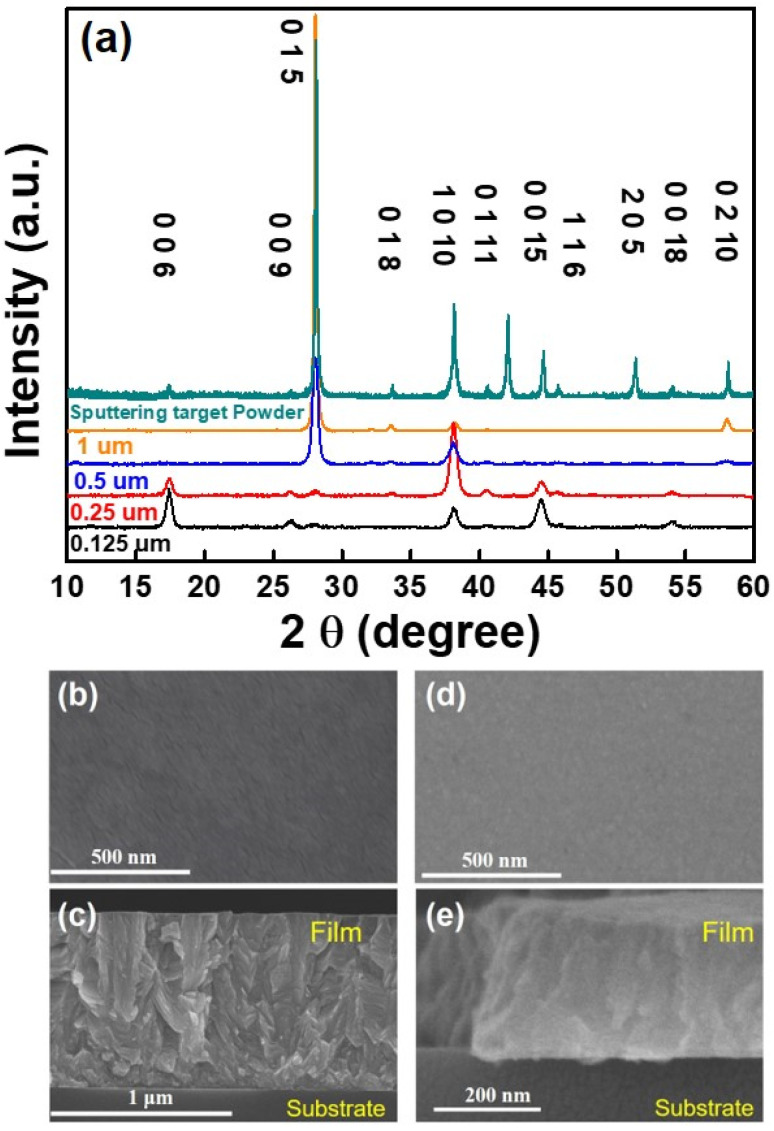
Film structure characterizations (**a**) Comparison of XRD patterns. (**b**) Top view of 1 µm film. (**c**) Cross-section view of 1 µm film. (**d**) Top view of 0.25 µm film. (**e**) Cross-section view of 0.25 µm film.

**Figure 2 nanomaterials-14-00747-f002:**
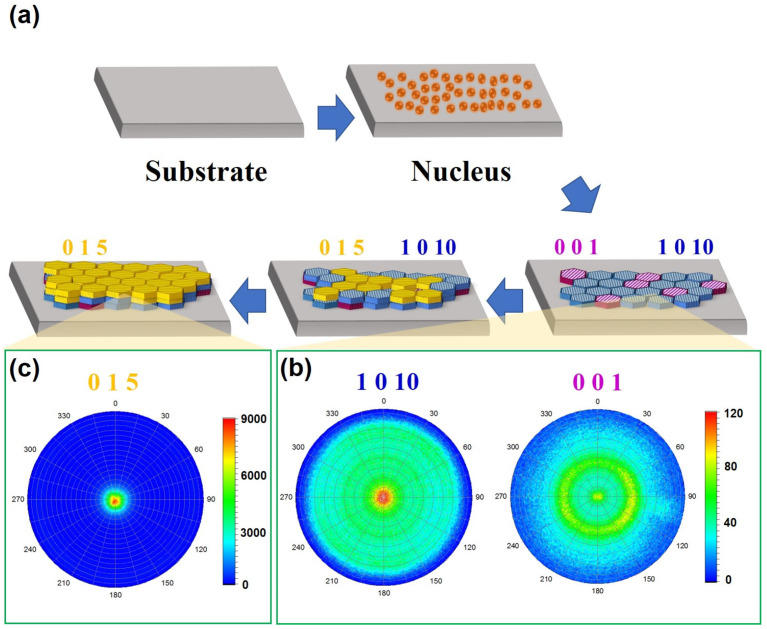
Magnetron sputtering-processed fabrication of highly textured Bi_0.5_Sb_1.5_Te_3_ thin films. (**a**) Schematic illustration of the crystal structure and growth process of Bi_0.5_Sb_1.5_Te_3_ thin films. (**b**) Pole figure of the 1 0 10 and 0 0 1 planes in the thin film with a thickness of 0.25 µm. (**c**) Pole figure of the 0 1 5 plane in the thin film with a thickness of 1 µm.

**Figure 3 nanomaterials-14-00747-f003:**
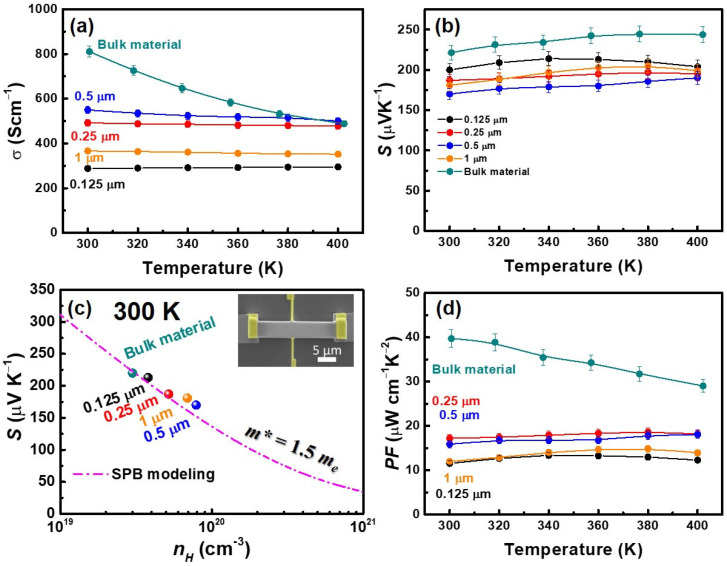
Thermoelectric transport properties of Bi_0.5_Sb_1.5_Te_3_. (**a**) Electrical conductivity. (**b**) Seebeck coefficient. (**c**) Room-temperature Pisarenko relationship with effective mass, *m** = 1.5 *m_e_*. (**d**) Power factor.

**Figure 4 nanomaterials-14-00747-f004:**
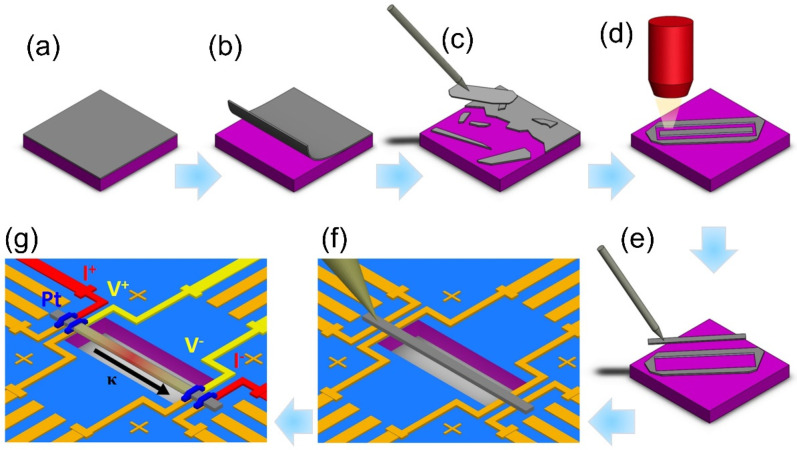
Sample preparation process for in-plane thermal conductivity measurements of films. (**a**) The thin film is grown on the substrate. (**b**) Peeling off the thin film. (**c**) Using a probe to transfer the peeled thin film onto the FIB cutting stage. (**d**) Precise cutting using the FIB. (**e**) Moving the cut sample onto the measurement chip. (**f**) Adjusting the position on the measurement chip. (**g**) Completion of electrode welding using FIB and preparation for measurement.

**Figure 5 nanomaterials-14-00747-f005:**
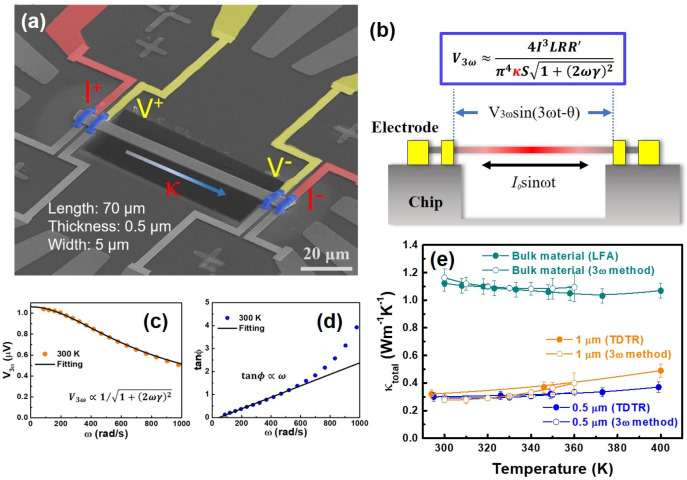
Thermal conductivity measurements. (**a**) The SEM image shows the sample suspended on the measurement platform and the application of four electrodes. (**b**) The illustration displays the four-probe configuration utilized to measure the thermal conductivity of a filament-like specimen. (**c**) Frequency dependence analysis of the third harmonic signal from the 0.5 µm thick film at 300 K is shown. (**d**) The relationship between tanϕ and frequency is illustrated. (**e**) Presents results from the 3ω method for 0.5 µm and 1 µm films, compared with results obtained using the 3ω method and LFA on the bulk material.

**Figure 6 nanomaterials-14-00747-f006:**
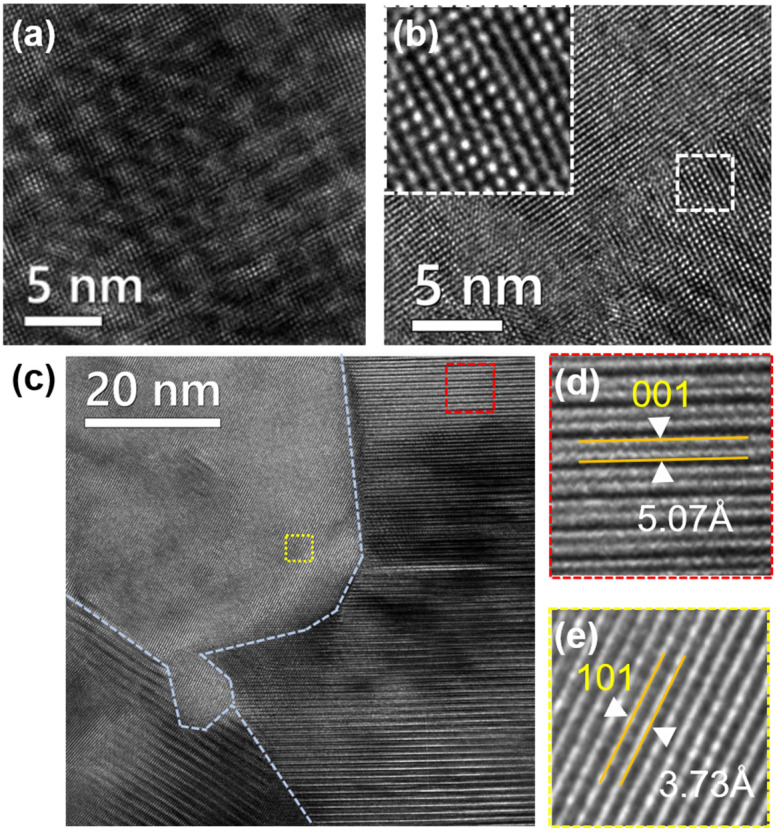
HRTEM analysis of the Bi_0.5_Sb_1.5_Te_3_ film, with a thickness of 0.25 µm. (**a**) Nano-moiré patterns extending across the area. (**b**) Enlarged view of the dotted white boxed region, where local defects are distributed randomly, appearing as dark spots. (**c**) A depiction of the area between grain boundaries. (**d**) Enlarged view of the dotted red boxed region in panel c. (**e**) Enlarged view of the dotted yellow boxed region in panel c.

**Figure 7 nanomaterials-14-00747-f007:**
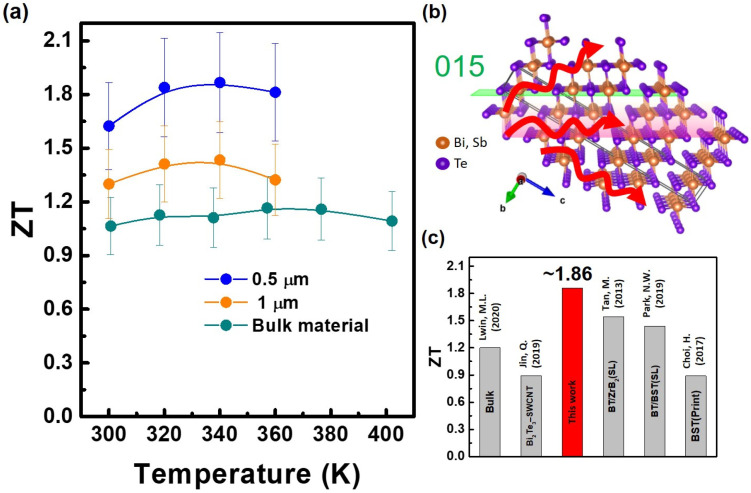
(**a**) ZT as a function of temperature. (**b**) Possible scattering scenarios of phonon propagation within the lattice structure. (**c**) ZT values for the advanced Bi_2_Te_3_-based bulk, films, BT/SWCNT nanocomposites, and SL films [7,28,29,30,31].

## Data Availability

The authors are unable to or have chosen not to specify which data has been used.

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
