# Peer review of "Enhancement of ZT in Bi0.5Sb1.5Te3 Thin Film through Lattice Orientation Management"

_nanomaterials, 2024, doi:10.3390/nano14090747_

Round 1

Reviewer 1 Report

Comments and Suggestions for Authors

Overview and general remarks:

The authors studied and discussed the improvement of thermoelectric properties of Bi-Sb-Te films in the relationships of their texture evolutions with increasing film thicknesses, via characterizations of their electrical properties and their thermal conductivity nature. Though they performed careful experiments, it seems that the contribution of annealing process was not discussed in detail.

I suppose this paper is worth publishing after major revisions.

1. Line 50 about expressions of anti-site defects

Host sites (elements) should be under-scripted.

2. Line 56 “equilibrium annealing“

A different term like “saturation annealing” was also used in this text, (Abstract and Conclusion part).

3. Line 73 “For instance, following thermal treatment, the PF of the 1 μm film increased from 299 to 1353 μWcm-1K-2,…”

It may be acceptable to be shown in the section of “Introduction”, but details should be explained in the section of “Experimental” and “Results and Discussion”. In Section 2.2, there explained shortly about conditions of annealing. However, there were no explanations about the effects of annealing to textural change, crystalline quality of Bi-Sb-Te films, or their electric/thermal properties. Rearrangement of text is strongly requested.

4. Line 105 “he glass was used…”

What kind of glass substrate was used in this experiment? Pure SiO2 glass? This issue should be notified, if possible.

If volatile elements like Na or B exist as additives in the glass body, I wonder whether they might cause some affection to Bi-Sb-Te films. But such may be  another issue to be discussed in future.

5. Line 119 “a triple-axis X-ray diffractometer”

The term “triple-axis (or triple-axis configuration)” for XRD measurements is commonly employed for XRD measurements using incident X-ray monochromators and receiving analyzer crystals to characterize crystalline quality of single crystalline epitaxial films/substrates.

6. Line 143

Some references to explain theoretical/technical basis of “The 3ω technique” should be cited.

7. Line 156 “a pronounced peak at (015)” and many others

   Diffraction indices should be shown WITHOUT brackets.

   Crystal faces or morphological planes should be shown WITH brackets.

This rule should be also applied to Fiure 1 (a), and the indices of Pole figure measurements, etc.

8. Line 158 “preferred orientation along the (015) direction”

I would like to ask your caution in case of using this kind of expression. As mentioned above, the term “(015)” is used to express the crystal faces or morphological planes of a crystal. So, the expression as “along the normal (perpendicular) direction of (015) planes” may be recommended.

And more importantly to notice, “[015] direction” and “the normal direction of (015) planes” are not parallel with each other if not the crystal structure is of cubic symmetry.

9. Line 165 “the proposed Stranski-Krastanov growth model”

If this is a proposed model, appropriate reference(s) should be cited. But I think the term “Stranski-Krastanov growth mode” is originally used for the epitaxial growth, and it cannot be easily extended to non-epitaxial cases.

10. Line 166 “During the initial stages of film growth,…” and following sections.

There were no comments for the effect of “annealing process” to the morphological or textural change in Bi-Sb-Te films. Figures 1 (b)-(c) are supposed to show the SEM views AFTER the annealing process. It means the resultant morphology and texture of films cannot be simply conclude as the result of deposition process. Comparison with characterization results of before/after annealing is required.

11. Line 170 “with a (00L)-oriented structure

This expression is wrong. This should be replaced as “with a (001)-oriented structure”.

12. Line 174 “along the (015) and (1010) orientations”

For the expression of index of (1010), I simply advice you to set spaces between numbers like 1 0 10, or to set underbars to the number of two (or more) digits, like 1010.

13. Line 203 “The growth of films typically goes through several stages” and to the end of this section

As mentioned in the comment 10, the effect of “annealing process” to the morphological or textural change in films should be discussed.

14. Line 217 “the development of 3D islands during the later stages of growth (Stranski-Krastanov mode)

Experimental evidences showing the occurrence of island growth mode are not clear.

15. Page 8, Figure 5(e) and corresponding part of text

The 3ω method was conducted only in the temperature range from 300 to 360 K. It is desirable (not required) to explain the experimental limitations, if any.

16. Line 326, “ofresistance change”

of resistance change

17. Figure 6,

Explanations of Figure 6 (a)-(c) are missing.

18. Line 400 “they likely encounter numerous boundaries as illustrated in Figure 7b” and the Figure 7(b),

Readers cannot easily understand from Figure 7(b) what “numerous boundaries” stand for in the relationship of crystallographic orientation of the films. I cannot be sure whether authors intended to emphasize numerous possibilities of phonon scattering at the inter-grain boundaries, but no clear correlation was clearly stated between textural evolutions, grain growth (especially growth along lateral direction), and thermal conductivity.

Author Response

We deeply appreciate the reviewers’ constructive comments, and have carefully revised our manuscript accordingly. These changes are included in the revised manuscript. The following are our responses to the reviewers’ individual comments.

Reviewer :

The authors studied and discussed the improvement of thermoelectric properties of Bi-Sb-Te films in the relationships of their texture evolutions with increasing film thicknesses, via characterizations of their electrical properties and their thermal conductivity nature. Though they performed careful experiments, it seems that the contribution of annealing process was not discussed in detail.

I suppose this paper is worth publishing after major revisions.

  1. Line 50 about expressions of anti-site defects

Host sites (elements) should be under-scripted.

Authors’ response:

We appreciate this kind suggestion. We have carefully reviewed your comments and made corresponding modifications to better express the anti-site defects. The new description is as follows:

“…….arise from anti-site defects like SbTe and BiTe [10,11].”

  1. Line 56 “equilibrium annealing” a different term like “saturation annealing” was also used in this text, (Abstract and Conclusion part).

Authors’ response:

Thank you for the valuable suggestion. We have made modifications according to the advice. All such annealing processes are uniformly expressed as " equilibrium annealing"

  1. Line 73 “For instance, following thermal treatment, the PF of the 1 μm film increased from 299 to 1353 μWcm-1K-2,…”

It may be acceptable to be shown in the section of “Introduction”, but details should be explained in the section of “Experimental” and “Results and Discussion”. In Section 2.2, there explained shortly about conditions of annealing. However, there were no explanations about the effects of annealing to textural change, crystalline quality of Bi-Sb-Te films, or their electric/thermal properties. Rearrangement of text is strongly requested.

Authors’ response:

Thanks for the kind remind. We have carefully checked and corrected the statements again.

  • The statement in Line 73, 'For instance, following thermal treatment, the PF of the 1 μm film increased from 299 to 1353 μWm-1K-2,' was incorrectly placed and needs to be removed.
  • The new statement is: "For instance, we achieve a notable enhancement of the Seebeck coefficient within the range of 170 to 220 μVK-1, optimizing the PF to a peak value of approximately 18.1 μWcm-1K-2."
  • During the sputtering process, the substrate was preheated to approximately 453 K. The structure remains almost identical before and after annealing. The most significant impact of annealing is its ability to effectively repair the anti-site defects in the films. Textural changes are mainly related to the thickness of the film, as explained by the "Stranski-Krastanov-like " model that we cited to describe the possible evolution of texture.

  1. Line 105 “he glass was used…”

What kind of glass substrate was used in this experiment? Pure SiO2 glass? This issue should be notified, if possible.

If volatile elements like Na or B exist as additives in the glass body, I wonder whether they might cause some affection to Bi-Sb-Te films. But such may be another issue to be discussed in future.

Authors’ response:

  • The Corning 1737F glass was used as a substrate in this experiment, and its surface roughness is in the range of 5 ~ 1.0 nm.
  • According to the composition report of the Corning 1737F glass, it is not purely SiO2. Although there are small amounts of B2O3 and trace amounts of Na2O, we believe that the effects of Na or B should be negligible in our sputtering process and heat treatment temperature range (~ 473 K). This has been confirmed by the EPMA analysis we conducted, which did not detect any Na or B elements.

We have included the model number of the glass substrate and the issues raised by the reviewer in Section 2-2.

  1. Line 119 “a triple-axis X-ray diffractometer”

The term “triple-axis (or triple-axis configuration)” for XRD measurements is commonly employed for XRD measurements using incident X-ray monochromators and receiving analyzer crystals to characterize crystalline quality of single crystalline epitaxial films/substrates.

Authors’ response:  

Our thin film exhibits textured features, meaning that the thin film structure has a preferred orientation. The preferred orientation is typically described in terms of pole figures, and data regarding this aspect must be obtained through “a multi-axis X-ray diffractometer”.

We have added the purpose of using “a multi-axis X-ray diffractometer” to Section 2.3.

  1. Line 143

Some references to explain theoretical/technical basis of “The 3ω technique” should be cited.

Authors’ response:

Thank you for the valuable suggestion. Two relevant references have been added to the revised manuscript.

[14] Lu, L.; Yi, W.; Zhang, D.L. 3 omega method for specific heat and thermal conductivity measurements. Rev. Sci. Instrum. 2001, 72, 2996-3003. https://doi.org/10.1063/1.1378340

[15] Li, G.; Liang, D.; Qiu, R.L.J.; Gao, X.P.A. Thermal conductivity measurement of individual Bi2Se3 nano-ribbon by self-heating three-ω method. Appl. Phys. Lett. 2013, 102, 043104. https://doi.org/10.1063/1.4789530

  1. Line 156 “a pronounced peak at (015)” and many others Diffraction indices should be shown WITHOUT brackets. Crystal faces or morphological planes should be shown WITH brackets. This rule should be also applied to Figure 1 (a), and the indices of Pole figure measurements, etc.

Authors’ response:

In both Figure 1(a) and the pole figure, "indices" are used to label crystal

faces.  In addition, Line 156 “a pronounced peak at (0 1 5)” has been revised as: “a pronounced peak observed at 2θ = 28.0° corresponds to the (0 1 5) crystal plane.”

  1. Line 158 “preferred orientation along the (015) direction” I would like to ask your caution in case of using this kind of expression. As mentioned above, the term “(015)” is used to express the crystal faces or morphological planes of a crystal. So, the expression as “along the normal (perpendicular) direction of (015) planes” may be recommended.

And more importantly to notice, “[015] direction” and “the normal direction of (015) planes” are not parallel with each other if not the crystal structure is of cubic symmetry.

Authors’ response:

Thank you for the valuable suggestion. Line 158 “preferred orientation along the (0 1 5) direction” has been revised as suggested: "preferred orientation along the normal direction of the (0 1 5) crystal plane."

  1. Line 165 “the proposed Stranski-Krastanov growth model” If this is a proposed model, appropriate reference(s) should be cited. But I think the term “Stranski-Krastanov growth mode” is originally used for the epitaxial growth, and it cannot be easily extended to non-epitaxial cases.

Authors’ response:

We appreciate this kind suggestion. We will follow the suggestion and modify the term to " Stranski-Krastanov-like " which may be more appropriate. Additionally, this term appearing in the abstract section will be replaced without specifically emphasizing such a model. An appropriate reference has been added in the revise manuscript.

[16] Tan, M.; Hao, Y.; Deng, Y.; et al. Tilt-structure and high-performance of hierarchical Bi1.5Sb0.5Te3 nanopillar arrays. Sci. Rep. 2018, 8, 6384. https://doi.org/10.1038/s41598-018-24872-4

  1. Line 166 “During the initial stages of film growth,…” and following sections. There were no comments for the effect of “annealing process” to the morphological or textural change in Bi-Sb-Te films. Figures 1 (b)-(c) are supposed to show the SEM views AFTER the annealing process. It means the resultant morphology and texture of films cannot be simply conclude as the result of deposition process. Comparison with characterization results of before/after annealing is required.

Authors’ response:

The thin films were prepared under conditions with the substrate heated to approximately 453 K. There was minimal change in XRD before and after heat treatment, indicating that the structure of the thin films was already fixed during the deposition process. Subsequent heat treatment at 473 K was only conducted to repair compositional defects. Figures 1(b)-(c) depict the results after heat treatment. We have already included these explanations in the Section 3.1 and the figure caption of Figure 1 in revise manuscript.

  1. Line 170 “with a (00L)-oriented structure”.

This expression is wrong. This should be replaced as “with a (001)-oriented structure”. 

Authors’ response:

We greatly appreciate this kind reminder. This expression has been corrected as“with a (0 0 l)-oriented structure”.

  1. Line 174 “along the (015) and (1010) orientations”

For the expression of index of (1010), I simply advice you to set spaces between numbers like 1 0 10, or to set underbars to the number of two (or more) digits, like 1010.

Authors’ response:

Thank you for the valuable suggestion. We have followed the advice to check and correct the indexes, for example, “along the (0 1 5) and (1 0 10) orientations”

  1. Line 203 “The growth of films typically goes through several stages” and to the end of this section

As mentioned in the comment 10, the effect of “annealing process” to the morphological or textural change in films should be discussed.

Authors’ response:

Following the response in comment 10, we have included all relevant supplementary explanations.

  1. Line 217 “the development of 3D islands during the later stages of growth (Stranski-Krastanov mode)”

Experimental evidences showing the occurrence of island growth mode are not clear.

Authors’ response:

We agree with the reviewer's comments. At this stage, we believe that such a growth model provides a better explanation for the evolution of thin film structures.

  1. Page 8, Figure 5(e) and corresponding part of text

The 3ω method was conducted only in the temperature range from 300 to 360 K. It is desirable (not required) to explain the experimental limitations, if any.

Authors’ response:

We attempted to raise the temperature measurement above 360 K; however, due to the non-linear variation of the sample's R', the measurement error became significant. This limitation is encountered in the current 3ω measurement of this sample. We have added this explanation to the revised version.

  1. Line 326, “ofresistance change”

of resistance change

Authors’ response:

This is a typographical error. We have corrected it.

  1. Figure 6,

Explanations of Figure 6 (a)-(c) are missing.

Authors’ response:

In addition to providing explanations for the original 6(a)~(c), we will also label and provide explanations for the two additional small images present in the original illustration, designated as (d) and (e) respectively.

The newly added explanations are as followings:

HRTEM analysis of the Bi0.5Sb1.5Te3 film, with a thickness of 0.25 µm. (a) Nano-moiré patterns extending across the area. (b) Local defects distributed randomly, appearing as dark spots. (c) A depiction of the area between grain boundaries. (d) Enlarged view of the dotted yellow boxed region in panel c. (e) Enlarged view of the dotted red boxed region in panel c.

Reviewer 2 Report

Comments and Suggestions for Authors

The paper presents improved TE performance of thin film Bi0.5Sb1.5Te3, which is one of the very hot materials for possible TE applications. The authors did very detailed studies for the microstructure and TE properties, and related each other. From the evolution of the preferred orientation with the thickness, they found the optimized thickness (which corresponds to the optimized orientations) for the better TE properties. The research is very interesting and I would like to recommend the publication. The only thing I would like to comment is the description in Line 22 "the development of the crystal structure...", I think the crystal structure is the same in all films as derived from the XRD data, but the microstructure (texture) changes with the growth thickness.   

Author Response

Response to reviewers

We deeply appreciate the reviewers’ constructive comments, and have carefully revised our manuscript accordingly. These changes are included in the revised manuscript. The following are our responses to the reviewers’ individual comments.

Reviewer:

The paper presents improved TE performance of thin film Bi0.5Sb1.5Te3, which is one of the very hot materials for possible TE applications. The authors did very detailed studies for the microstructure and TE properties, and related each other. From the evolution of the preferred orientation with the thickness, they found the optimized thickness (which corresponds to the optimized orientations) for the better TE properties. The research is very interesting and I would like to recommend the publication. The only thing I would like to comment is the description in Line 22 "the development of the crystal structure...", I think the crystal structure is the same in all films as derived from the XRD data, but the microstructure (texture) changes with the growth thickness.  

Authors’ response:

We greatly appreciate the positive comment.

In Line 22 "the development of the crystal structure...", has been corrected as " We show that the microstructure (texture) changes with the growth thickness in the films highly correlate with the thickness and can be explained by the Stranski-Krastanov-like model ".

Reviewer 3 Report

Comments and Suggestions for Authors

This is an interesting manuscript connected with thermoelectrical properties of thin films of Bi0.5Sb1.5Te3. But there are some week points which should be improved or explained They are listed below:

1. The following sentence in lines 73-76 is incomprehensible "For instance, following thermal treatment, the PF of the 1 μm film increased from 299 to 1353 μWcm-1K-2, marking an approximate 450% growth. We achieve a no-table enhancement of the Seebeck coefficient within the range of 170 to 220 μVK-1, optimizing the PF to a peak value of approximately 18.1 μWcm-1K-2.". Please explain what was the real value of Power Factor.

2. In subsections 2.1 and 2.2 we have two different descriptions of preparing the BST target for sputtering. Which one is true? Is this the original recipe of the Authors of the manuscript, or is it based on literature?

3. In line 105 there is incomprehensible sentence "he glass was used as a substrate with 20 x 20 mm2 dimension ...". What kind of glass was used as a substrate? What was its roughness?

4. The electrical and thermal properties of BST thin films were measured in very limited temperature range - between 300 and 400 K. Why?

5. Please explain (please give proper formula) for "sputtering target" presented in Figs. 3,  5 and 6.

6. Plese write the formula for SPB (single parabolic band) model mentioned in line 261. Please give proper reference for this model.

7. The measurementts of electrical conductivity and Seebeck coefficient were made for BST films deposited on glass substrates. A very clever measurement of the thermal conductivity of these films was made for pure BST films (without substrate). But in the case of practical use of such films, e.g. as arms of thermoelectric microgenerators, they will be applied on appropriate substrates. I would like to ask Authors for explanation how the thermal conductivity of the substrate will affect the operational parameters of the thermoelectric microgenerator containing thin-film of BST?

Based on above remarks I propose major revision of this manuscript.

Comments on the Quality of English Language

The language requires careful correction. Moreover, in the article the pressure should be expressed in "Pa" (not in "torr"), and the Power Factor (PF) in "microWm-1K-2" (not in "microWcm-1K-2").

Author Response

Response to reviewers

We deeply appreciate the reviewers’ constructive comments, and have carefully revised our manuscript accordingly. These changes are included in the revised manuscript. The following are our responses to the reviewers’ individual comments.

Reviewer:

This is an interesting manuscript connected with thermoelectrical properties of thin films of Bi0.5Sb1.5Te3. But there are some week points which should be improved or explained They are listed below:

  1. The following sentence in lines 73-76 is incomprehensible "For instance, following thermal treatment, the PF of the 1 μm film increased from 299 to 1353 μWcm-1K-2, marking an approximate 450% growth. We achieve a no-table enhancement of the Seebeck coefficient within the range of 170 to 220 μVK-1, optimizing the PF to a peak value of approximately 18.1 μWcm-1K-2.". Please explain what was the real value of Power Factor.

Authors’ response:

Thank you for the kind remind. We have carefully checked and corrected the statements again. The new statement is:

"For instance, we achieve a notable enhancement of the Seebeck coefficient within the range of 170 to 220 μVK-1, optimizing the PF to a peak value of approximately 18.1 μWcm-1K-2."

  1. In subsections 2.1 and 2.2 we have two different descriptions of preparing the BST target for sputtering. Which one is true? Is this the original recipe of the Authors of the manuscript, or is it based on literature?

Authors’ response:

  The descriptions of preparing the BST target for sputtering in Subsections 2.1 is true. We have re-edited it in the revised manuscript.

  1. In line 105 there is incomprehensible sentence "he glass was used as a

substrate with 20 x 20 mm2 dimension ...". What kind of glass was

used as a substrate? What was its roughness?

Authors’ response:

  • The sentence you pointed out is indeed a typo. The correct sentence should be: "The glass was used as a substrate with 20 x 20 mm2 dimension ...".
  • The Corning 1737F glass was used as a substrate, and its surface roughness Ra is in the range of 5 ~ 1.0 nm.

  1. The electrical and thermal properties of BST thin films were measured in very limited temperature range - between 300 and 400 K. Why?

Authors’ response:

The measurement of electrical conductivity and Seebeck coefficient of the thin film up to 473 K is achievable. However, it has been observed that its peak power factor (PF) typically falls between 360 K to 400 K. Additionally, the measurement of the thermal conductivity of the suspended thin film is further constrained by experimental conditions, allowing measurement only up to 360 K. Therefore, reporting the electrical and thermal properties in the range of 300 K to 400 K in this article is sufficient to provide an overview of the thermoelectric properties of these films.

  1. Please explain (please give proper formula) for "sputtering target" presented in Figs. 3, 5 and 6.

Authors’ response:

  In Figures 3, 5, and 7, the term "sputtering target" refers to the bulk material used for comparison with the thin films in terms of their thermoelectric properties. All thin films are obtained through the sputtering process using this bulk material. To avoid confusion, we will replace " sputtering target " with "Bulk material" in Figures 3, 5, and 6, and provide definitions for this term in the experimental section Sec. 2.1.

  1. Please write the formula for SPB (single parabolic band) model mentioned in line 261. Please give proper reference for this model

Authors’ response:

Thank you for the valuable suggestion. The formula for SPB (single parabolic band) model and the relevant reference have been added in the revise manuscript.

(1)                                                  

(2)                                                              

(3)  

(4)                                                

"The above formula can be seen in word"

where the Hall carrier density, denoted as nH​, is linked to the chemical carrier density n through the equation n via nH = n/rH ​, where rH represents the Hall factor for acoustic phonon scattering. The function Fx(η) denotes the x-th order Fermi integral.

[19]  Pei, Y.; LaLonde, A. D.; Wang, H. & Snyder, G. J. Low effective mass leading to high thermoelectric performance. Energy Environ. Sci. 2012, 5, 7963-7969. https://doi.org/10.1039/C2EE21536E                                                                                                     

  1. The measurements of electrical conductivity and Seebeck coefficient were made for BST films deposited on glass substrates. A very clever measurement of the thermal conductivity of these films was made for pure BST films (without substrate). But in the case of practical use of such films, e.g. as arms of thermoelectric microgenerators, they will be applied on appropriate substrates. I would like to ask Authors for explanation how the thermal conductivity of the substrate will affect the operational parameters of the thermoelectric microgenerator containing thin-film of BST?

Authors’ response:

The thermal conductivity of the substrate can significantly affect the operational parameters of a thermoelectric microgenerator containing a thin film of BST. Possible reasons are as following:

  • Thermal Losses: A substrate with high thermal conductivity can mitigate thermal losses by efficiently conducting heat away from the device, reducing parasitic heat loss that doesn't contribute to power generation.
  • Temperature Gradient: The temperature difference across the thermoelectric material is crucial for generating electricity via the Seebeck effect. A substrate with high thermal conductivity can decrease a larger temperature gradient between the hot and cold sides of the device, thereby reducing the power output.

In summary, the thermal conductivity of the substrate plays a critical role in determining the efficiency, power output, and reliability of thermoelectric microgenerators containing thin films of BST.

Round 2

Reviewer 1 Report

Comments and Suggestions for Authors

Overview and general remarks:

The authors revised and improved their manuscript well, but some more minor revisions are requested, as followed. I suppose this paper is worth publishing after those minor revisions.

1. Line 112 “a multi-axis X-ray diffractometer”

In general, this kind of XRD system, like MRD apparatus, is called as “a four-circle diffractometer”, where a tilting axis (“chi” or “psi”) and a rotating axis around surface normal (“phi”) are installed together with the omega axis and the 2theta axis.

2. Line 177 “(006) pole figure“ and Authors’ response to my comment No.7

Diffraction indices should be shown WITHOUT brackets. This is the rule in the field of crystallography.

Revisions for Figure 1(a), and Figure 2 (a),(b),(c) and for the expression in Line 177 are required.

For example, Figure 2(b) right is the data of Pole figure measurement for 006 reflection. It can be interpreted to show the distribution of poles of (001) plane. So, why did you label only for (006), instead of (001), (002), (004) or others?

In the same course of discussion, it is a very strange expression to use “(0 0 6)” in the right-down panel of Figure 2(a). Why did you pick-up only for (006), instead of (001), (002), (004) or others? Here should be used “(0 0 1)”.

3. Line 410 “they likely encounter numerous boundaries as illustrated in Figure 7b, leading to substantial scattering,”

I am not sure that I have previously commented as “issue 18”,

“Readers cannot easily understand from Figure 7(b) what “numerous boundaries” stand for in the relationship of crystallographic orientation of the films. I cannot be sure whether authors intended to emphasize numerous possibilities of phonon scattering at the inter-grain boundaries, but no correlation was clearly stated between textural evolutions, grain growth (especially growth along lateral direction), and thermal conductivity.”

I wonder authors discussed with images of pillar-like (or columnar) growth textures in the BST film, but it is not clearly explained in the text.

Or, I guess authors intended to explain under the correlations with the moire patterns shown in Figure 6(a) or (b), but still it is unclear.

5. Line 416 “dotted yellow boxed region” -> “dotted red boxed region”

6. Line 417 “dotted red boxed region” -> “dotted yellow boxed region”

7. Line 427 “on Bi2Te3-based alloy thin films” -> numbers should be under-scripted

Following comments are intended to give some advices for future work of authors, not requesting manuscript revision.

<about “Stranski-Krastanov-like” growth mode>

Still, I cannot agree to use this term for the present cases, since these films were not epitaxial ones nor grown on single-crystalline substrates. But, I understand that the proposal to apply this term was done with other researchers.

I suppose that the initial growth layer “without a distinct growth direction” (Line 157) adjacent to glass substrates is just a wetting buffer layer, with no correlation to the single-crystalline layer part explained in S-K mode growth.

<about the effect of the equilibrium annealing>

In the “Response to reviewer” letter, you explained as “The structure remains almost identical before and after annealing”, but this FACT was not clearly shown in the manuscript. I think it important to show experimental results as the basis to reach to this judgment. XRD profiles before/after annealing may be useful.

<about the Pole figure data of 0 1 5 reflection shown in Figure 2(c)>

I wonder there should be observed some rings apart by 50, 85, 63 (=180-117) degrees from the center of the pole figure data, like rings observed in Figure 2(b).

Author Response

Response to reviewers

We deeply appreciate the reviewers’ constructive comments, and have carefully revised our manuscript accordingly. These changes are included in the revised manuscript. The following are our responses to the reviewers’ individual comments.

Reviewer :

  1. Line 112 “a multi-axis X-ray diffractometer”

In general, this kind of XRD system, like MRD apparatus, is called as “a four-circle diffractometer”, where a tilting axis (“chi” or “psi”) and a rotating axis around surface normal (“phi”) are installed together with the omega axis and the 2theta axis.

Authors’ response:  

We have corrected the wording as suggested to " a four-circle diffractometer " in the revised manuscript.

  1. Line 177 “(006) pole figure“ and Authors’ response to my comment No.7

Diffraction indices should be shown WITHOUT brackets. This is the rule in the field of crystallography.

Revisions for Figure 1(a), and Figure 2 (a),(b),(c) and for the expression in Line 177 are required.

For example, Figure 2(b) right is the data of Pole figure measurement for 006 reflection. It can be interpreted to show the distribution of poles of (001) plane. So, why did you label only for (006), instead of (001), (002), (004) or others?

In the same course of discussion, it is a very strange expression to use “(0 0 6)” in the right-down panel of Figure 2(a). Why did you pick-up only for (006), instead of (001), (002), (004) or others? Here should be used “(0 0 1)”.

Authors’ response:

Thank you for the valuable suggestion. We have followed the advice to correct the brackets, for example, "0 1 5, 1 0 10 and 0 0 1”, in the revised manuscript.

  1. Line 410 “they likely encounter numerous boundaries as illustrated in Figure 7b, leading to substantial scattering,”

I am not sure that I have previously commented as “issue 18”,

“Readers cannot easily understand from Figure 7(b) what “numerous boundaries” stand for in the relationship of crystallographic orientation of the films. I cannot be sure whether authors intended to emphasize numerous possibilities of phonon scattering at the inter-grain boundaries, but no correlation was clearly stated between textural evolutions, grain growth (especially growth along lateral direction), and thermal conductivity.”

Authors’ response:
The preferred orientation of the thin films discussed in the text is 0 1 5. Therefore, Figure 7(b) is specifically emphasized and elaborated to illustrate the correlation between the scattering process of phonons within the lattice and the predominant 0 1 5 orientation.

I wonder authors discussed with images of pillar-like (or columnar) growth textures in the BST film, but it is not clearly explained in the text.

Authors’ response:

The evolution of pillar-like (or columnar) growth textures in BST thin films as depicted in Figure 2 has been clearly elucidated through structural identification data in the text .

Or, I guess authors intended to explain under the correlations with the moire patterns shown in Figure 6(a) or (b), but still it is unclear.

Authors’ response:

We appreciate this kind suggestion. To facilitate readers' understanding of the meaning of Figure 7(b), modifications will be made to Line 409 to Line 412 “From the results of HRTEM, …….around 0.3 Wm-1K-1 .”

The revised statement is:

“In addition to the complex interface-enhanced phonon scattering observed in HRTEM in Figure 6, the low thermal conductivity of these thin films is also attributed to their textured structure, with a preferred crystal orientation of 015. Therefore, by using the schematic diagram in Figure 7b, one can further understand the possible scattering scenarios of phonons within grains in addition to interface scattering. Considering these factors collectively contributes to achieving a thermal conductivity of approximately 0.3 Wm-1K-1.”

  1. Line 416 “dotted yellow boxed region” -> “dotted red boxed region”

Authors’ response:

We have followed the advice to correct the“dotted red boxed region”

  1. Line 417 “dotted red boxed region” -> “dotted yellow boxed region”

Authors’ response:

We have followed the advice to correct the“dotted yellow boxed region”

  1. Line 427 “on Bi2Te3-based alloy thin films” -> numbers should be under-scripted

Authors’ response:

We have corrected the term to“Bi2Te3-based” according to the suggestion.

Reviewer 3 Report

Comments and Suggestions for Authors

Once again I would like to confirm that this is an interesting manuscript connected with thermoelectrical properties of thinfilms of Bi0.5Sb1.5Te3.

All my suggestions were taken into account and I propose to publish thi s paper as is. Only on p. 9 and 10 it is necessary to changr numbers of Equations - they should be 5, 6 and 7 (not 1,2,3).

Author Response

Response to reviewers

We deeply appreciate the reviewers’ constructive comments, and have carefully revised our manuscript accordingly. These changes are included in the revised manuscript. The following are our responses to the reviewers’ individual comments.

Reviewer :

All my suggestions were taken into account and I propose to publish this paper as is. Only on p. 9 and 10 it is necessary to change numbers of Equations - they should be 5, 6 and 7 (not 1,2,3).

Authors’ response:

We have carefully sorted the numbering of these equations correctly.
